# Dietary supplementation of alpha-lipoic acid mitigates the negative effects of heat stress in broilers

**Sanjeev Wasti, Nirvay Sah, Chin N. Lee, Rajesh Jha, Birendra Mishra***

Department of Human Nutrition Food and Animal Sciences, University of Hawaii at Manoa, Honolulu, HI, United States of America

* bmishra@hawaii.edu

**Data Availability Statement:** Additional supplemental files having the minimal data set is submitted along with this manuscript.

## Abstract

Heat stress accounts for substantial economic loss in the poultry industry by altering the health and performance of chickens. Alpha-lipoic acid (ALA) is a water and fat-soluble antioxidant which is readily absorbed from the intestine resulting in maximum bioavailability. Moreover, ALA acts as a coenzyme in glucose metabolism and helps generate other antioxidants. Considering these benefits, we hypothesized that dietary supplementation of ALA would help mitigate heat stress in poultry. A total of 72 Day-old broiler chicks were randomly assigned into three treatment groups: no heat stress (NHS), heat stress with basal diet (HS), and heat stress with alpha-lipoic acid (HS+ALA); each treatment group had 6 replicate pens with 4 birds in each pen (n = 24/group). The allocated birds were raised under standard husbandry practices for 3 weeks. After 21 d, birds in the HS and HS+ALA groups were exposed to heat stress (33°C for 8 hours during the day) for 3 weeks, while the NHS group was reared under normal conditions (22–24°C). The HS+ALA group received a basal finisher diet fortified with ALA (500 mg/kg) during the treatment period (22 to 42 d), while other birds were provided with the basal finisher diet. Weekly body weight and feed intake were recorded. The cecum digesta for volatile fatty acids (VFAs) analysis and 16S rRNA sequencing for the gut microbiota analysis; and the ileum tissue samples for histological and gene expression analyses were collected on d 42. Exposure to heat stress decreased (P<0.05) average daily gain (ADG) and final body weight (FBW) in the HS group compared to the NHS group, the supplementation of ALA improved (P<0.05) ADG and FBW in heat-stressed birds. Furthermore, birds in the HS+ALA group had increased (P<0.05) expression of *HSP90*, *PRDX1*, *GPX3*, *SOD2*, *OCLN*, and *MUC2* genes and higher (P<0.05) concentrations of major VFAs (acetate, propionate, and butyrate). The dietary ALA supplementation also improved the villus height and villus height to crypt depth ratio in the HS+ALA group. The microbial diversity analysis revealed significant abundance (P<0.05) of beneficial bacteria *Lactobacillus* and *Peptostreptococcaceae* in the cecum of the ALA group. These results indicate that dietary ALA supplementation effectively mitigates the negative effects of heat stress in broilers by improving the expression of heat-shock, tight-junction, antioxidants, and immune-related genes in the intestine, improving villus structures, increasing concentration of major VFAs, and enriching the beneficial microbiota.

**Funding:** This work was supported by a Start-up grant from CTAHR University of Hawaii at Manoa, and USDA Multistate (2052R) to B.M.

**Competing interests:** The authors declare that they have no competing interests.

## Introduction

Poultry meat is a good source of white meat containing a low amount of fat and high proteins. Considering this benefit of poultry meat, the consumers' preference for poultry meat has rapidly increased in the past decade. This demand has been largely fulfilled by intensive selection and breeding of the chickens for rapid growth and heavier breast muscle [1]. However, these improved breeding are accompanied by one major problem, i.e., highly susceptible to higher environmental temperature due to higher metabolic rate, lack of sweat glands, and the presence of feathers [2]. Thus, with the rising global temperature, heat stress is a significant problem in the poultry industry resulting in massive economic loss in the tropical regions [3]. Heat stress exhibits negative effects on poultry health and production performance by altering different physiological, neuroendocrine, immunological, and behavior responses [2]. As a result, there has been a huge surge in the research regarding mitigating the heat stress in poultry. Researchers have recently tried and tested different compounds such as polyphenols and antioxidants to ameliorate the heat stress in poultry [2, 4–7].

Alpha-lipoic acid (ALA), also known as Universal antioxidant, is produced in small amounts inside the cell [8]. ALA is both water and fat-soluble antioxidants, are readily absorbed from the intestine, and can easily cross the blood-brain barrier resulting in optimum bioavailability. Besides this, both ALA and dihydrolipoic acid (DHLA)- the reduced form of ALA, can quench free radicals both in the liquid and aqueous domains [9]. Moreover, ALA and DHLA also possess metal-chelating activity, acts as a coenzyme in glucose metabolism, and generates other antioxidants such as ascorbate, vitamin E, and glutathione (GST) [8, 10]. Considering these benefits, ALA has recently gained huge attention in normal and heat stress conditions as a potential feed supplement in poultry.

Studies have shown that dietary fortification of ALA was able to improve the growth performance indices, immunological and biochemical characteristics, lipid metabolism, and oxidative stress in the poultry [9]. Likewise, the storability of poultry meat and meat product was also found to increase with the fortification of ALA in the feed [9]. However, there has not been any study focusing on ALA supplementation on gastrointestinal physiology and the growth performance in heat-stressed broiler birds. Thus, this study was carried out to unveil ALA supplementation's mitigatory effects on the growth performance, immune parameters, gut microbiota, and gut health of heat-stressed broiler birds.

## Material and methods

### Birds and husbandry

The animal experimentation was carried out at the small animal facility of the University of Hawaii at Manoa, and the experimental protocol was approved by the Institutional Animal Care and Use Committee of the University of Hawaii (Approval no. 17–2605). A total of 72 day-old Cobb-500 unsexed chicks were sourced from a local hatchery (Asagi Hatchery Inc., Honolulu, HI). On d 1, birds were weighed individually, winged tagged, and placed equally and randomly to one of 3 treatments with 6 replicates of each treatment (n = 24 birds/treatment; 4 birds/pen). The treatment groups were: 1) No heat stress (NHS), 2) Heat stress with basal diet (HS), and 3) heat stress with alpha-lipoic acid (HS+ALA). For the first 21 d, all the birds were raised following the standard broiler rearing guidelines on the floor pen system with access to *ad libitum* feed and water. After 21 d, birds in the HS, and HS+ALA were exposed to the heat of 33–35˚C (8 am to 6 pm) and 21–22˚C (6 pm to 8 am) with 50% relative humidity for 3 weeks. Birds in the NHS group were reared at the normal room temperature

(22˚C-24˚C) with 50% relative humidity. The lighting regime was 23 h light and 1 h dark periods. The size of the pen was 1 m x 0.71 m, and the stocking density was 1500 cm$^2$/bird.

## Diets

Birds were fed the corn-soybean meal-based mash diets in two phases, starter (1–21 d) and finisher (22–42 d), to meet the nutrients requirements of broilers [11]. The energy and protein requirements of the diet was met following the NRC (1994); however, Ca and P level was lower than the commercial requirements. All the birds were provided with the normal starter diet for the first 14 d; afterward, 500 mg/kg ALA was supplemented on the starter diet of the HS+ALA group from 14 to 21 d, while the other two groups (HS and NHS) were provided with the normal starter diet. From 22 to 42 d, NHS and HS birds were fed with the basal finisher diets, and 500 mg/kg ALA was supplemented in the finisher diet of the HS+ALA group. The dose rate of ALA was considered as 500 mg/kg feed based on a previous study [12] in the broiler birds. The diet formulation and their nutrient profiles are presented in Table 1.

## Growth performance

The birds were individually weighed on d 1, 7, 14, 21, 28, 35, and 42. Feed provided to the individual pen was noted, and leftover feed in the pen was also weighed weekly. The feed intake per replicate pen was determined by subtracting the leftover feed over to the total feed consumed during the week. The mortality of the birds was monitored daily. The ADG, ADFI, and FCR were calculated after adjusting for mortality, if any.

## Sample collection

At 42 d, two birds/pen (n = 12) from each treatment group were selected randomly and were euthanized by carbon dioxide asphyxiation. A small piece of the ileum (5 cm posterior to the ileocecal transition) was collected (n = 6/treatment; one from each pen), snap-frozen in liquid nitrogen, and stored at -80˚C for subsequent gene expression study. The cecum was excised, snap-frozen, and placed at -80˚C for microbiota (n = 6/treatment; one from each pen) and VFA (n = 12/treatment; two from each pen) analysis while, approximately 1 cm of the ileum sample (6 cm posterior to the ileocecal junction) was excised, washed with normal saline and preserved in 10% neutral buffered formalin (NBF, pH 7) for the ileum histomorphology analysis (n = 4/treatment).

## Total RNA extraction

The 50–100 mg of frozen ileum tissues were used for the total RNA extraction. The total RNA was extracted using TRIzol reagent (Invitrogen, Carlsbad, CA) following the manufacturer's instructions. The RNA concentration was determined using NanoDrop one (Thermo Fisher Scientific, Madison, WI), while quality was accessed by running RNA samples on 2% agarose gel. Extracted RNA was stored at -80˚C until further analysis.

## Quantitative real-time PCR (qPCR) assay

The specific oligonucleotide primers used for the qPCR assay were designed using the NCBI Primer-Blast tool and are presented in Table 2. The gene expression analysis was carried out as previously described [13]. In short, for cDNA synthesis, 1 μg of total RNA (20 μL reaction of reverse transcriptase mixture) was reverse transcribed using a High-Capacity cDNA Reverse Transcription Kit (Applied Biosystems, Foster City, CA). The newly synthesized cDNA (20 μL) was diluted (25X) with 480 μL of nuclease-free water and was stored at -20˚C until

**Table 1. Ingredients and nutrient composition of the experimental diets.**

| Ingredients % | Starter | | Finisher | |
| --- | --- | --- | --- | --- |
| | Basal | Basal+ALA | Basal | Basal+ ALA |
| Corn | 54.86 | 54.86 | 63.14 | 63.14 |
| SBM | 39.5 | 39.5 | 29.6 | 29.6 |
| Soybean oil | 2 | 2 | 4.5 | 4.5 |
| Limestone | 1.27 | 1.27 | 0.85 | 0.85 |
| Monocalcium phosphate | 0.75 | 0.75 | 0.5 | 0.5 |
| Lysine | 0.23 | 0.23 | 0.18 | 0.18 |
| Methionine | 0.14 | 0.14 | 0.12 | 0.12 |
| Threonine | 0.2 | 0.2 | 0.16 | 0.16 |
| NaCl | 0.43 | 0.43 | 0.35 | 0.35 |
| Sodium bicarbonate | 0.12 | 0.12 | 0.1 | 0.1 |
| Vitamin + Mineral mix[*] | 0.5 | 0.5 | 0.5 | 0.5 |
| ALA, mg/kg (top dressing) | 0 | 500 | 0 | 500 |
| Calculated nutrient content, % | | | | |
| MEn, kcal/kg | 2909 | 2909 | 3203 | 3203 |
| CP | 22.09 | 22.09 | 18.07 | 18.07 |
| Ca | 0.75 | 0.75 | 0.52 | 0.52 |
| Total P | 0.57 | 0.57 | 0.47 | 0.47 |
| Available P | 0.3 | 0.3 | 0.23 | 0.23 |
| Lysine | 1.39 | 1.39 | 1.10 | 1.10 |
| Methionine | 0.48 | 0.48 | 0.41 | 0.41 |
| Cystine | 0.43 | 0.43 | 0.38 | 0.38 |
| Threonine | 1.03 | 1.03 | 0.85 | 0.85 |
| Tryptophan | 0.33 | 0.33 | 0.26 | 0.26 |
| Methionine + Cysteine | 0.91 | 0.91 | 0.8 | 0.8 |
| Arginine | 1.61 | 1.61 | 1.31 | 1.31 |
| Valine | 1.22 | 1.22 | 1.03 | 1.03 |
| Isoleucine | 0.93 | 0.93 | 0.76 | 0.76 |
| Leucine | 1.89 | 1.89 | 1.63 | 1.63 |
| dig Lys | 1.25 | 1.25 | 0.99 | 0.99 |
| dig Met | 0.45 | 0.45 | 0.39 | 0.39 |
| dig Thr | 0.85 | 0.85 | 0.69 | 0.69 |
| NDF | 9.13 | 9.13 | 8.78 | 8.78 |
| CF | 3.97 | 3.97 | 3.46 | 3.46 |
| Na | 0.22 | 0.22 | 0.18 | 0.18 |
| Cl | 0.30 | 0.30 | 0.25 | 0.25 |
| Choline (mg/kg) | 1419 | 1419 | 1200 | 1200 |

[*]Providing the following (per kg of diet): vitamin A (trans-retinyl acetate), 10,000 IU; vitamin $D_3$ (cholecalciferol), 3,000 IU; vitamin E (all-rac-tocopherol-acetate), 30 mg; vitamin $B_1$, 2 mg; vitamin $B_2$, 8 mg; vitamin $B_6$, 4 mg; vitamin $B_{12}$ (cyanocobalamin), 0.025 mg; vitamin $K_3$ (bisulphatemenadione complex), 3mg; choline (choline chloride), 250 mg; nicotinic acid, 60 mg; pantothenic acid (D-calcium pantothenate), 15 mg; folic acid, 1.5 mg; betaíne anhydrous, 80 mg; D-biotin, 0.15 mg; zinc (ZnO), 80 mg; manganese (MnO), 70 mg; iron ($FeCO_3$), 60 mg; copper ($CuSO_4 \cdot 5H_2O$), 8 mg; iodine (KI), 2 mg; selenium ($Na_2SeO_3$), 0.2 mg.

further analysis. The qPCR was carried out by using StepOne Plus real-time PCR system (Applied Biosystems, Foster City, CA) where10 μL reaction mixture containing 3 μL of cDNA, 5 μL of PowerUp SYBR Green Master Mix (Applied Biosystems, Foster City, CA), and 1 μL each of forward and reverse primers specific to the gene target was used. The amplification

**Table 2. Primers used to quantify the expression of the genes by qPCR.**

| Gene | Accession no. | Primer Sequence | Amplicon (bp) |
|---|---|---|---|
| SOD1 | NM_205064.1 | F: CAACACAAATGGGTGTACCA | 119 |
| | | R: CTCCCTTTGCAGTCACATTG | |
| SOD2 | NM_204211.1 | F: CCTTCGCAAACTTCAAGGAG | 160 |
| | | R: AGCAATGGAATGAGACCTGT | |
| GPX1 | NM_001277853.2 | F: AATTCGGGCACCAGGAGAA | 101 |
| | | R: CTCGAACATGGTGAAGTTGG | |
| GPX3 | NM_001163232.2 | F: GAGGGAGAAGGTGAAATGCT | 192 |
| | | R: CCCAGCTCATTTTGTAGTGC | |
| TXN | NM_205453.1 | F: GGCAATCTGGCTGATTTTGA | 79 |
| | | R: ACCATGTGGCAGAGAAATCA | |
| PRDX1 | NM_001271932.1 | F: GGTATTGCATACAGGGGTCT | 101 |
| | | R: AGGGTCTCATCAACAGAACG | |
| NRF2 | NM_205117.1 | F: CCCTGCCCTTAGAGATTAGAC | 248 |
| | | R: CAAGTTCATGTCCTTTTCTCTGC | |
| HSF1 | NM_001305256.1 | F: AAGGAGGTGCTCCCAAAGTA | 221 |
| | | R: TTCTTTATGCTGGACACGCTG | |
| HSF3 | NM_001305041.1 | F: TTCAGCGATGTGTTTAACCCT | 244 |
| | | R: GGAGGTCTTTTGGATCCTCT | |
| HSP90 | NM_001109785.1 | F: GATAACGGTGAACCTTTGGG | 120 |
| | | R: GGGTAGCCAATGAACTGAGA | |
| HSP70 | NM_001006685.1 | F: TCTCATCAAGCGTAACACCAC | 104 |
| | | R: TCTCACCTTCATACACCTGGAC | |
| OCLN | NM_205128 | F: CCGAGGACAGCCCTCAATAC | 82 |
| | | R: CTTTGGTAGTCTGGGCTCCG | |
| CLDN1 | NM_001013611 | F: TACCCCAAAAATGCCCCCTC | 109 |
| | | R: GCGGCATTGTAGTGTCCTCT | |
| MUC2 | NM_001318434 | F: GTGGTCTGTGTGGCAACTT | 71 |
| | | R: GTCTCTTGCAGCCCATTCCT | |
| IL4 | NM_001030693 | F: TGTGCCCACGCTGTGCTTACA | 155 |
| | | R: CTTGTGGCAGTGCTGGCTCTCC | |

*F = Forward; R = Reverse.

conditions were 50˚C for 2 minutes (hold), 95˚C for 2 minutes (hold), followed by 40 repeat cycles of 95˚C for 15 seconds (denaturation), 60˚C for 15 seconds (annealing), and 72˚C for 1 minute (extension). Based on the consistency of expression in the ileum tissue, β-actin was chosen for normalization. The target genes were analyzed in duplicates, and an average value was taken from the experimental replicate. The expression level of target genes was determined using the cycle threshold (Ct) values and changes in the gene expression were calculated by the $2^{-\Delta\Delta Ct}$ method compared to the control group. The relative mRNA expression was normalized to the endogenous reference gene *B-actin*.

## Ileum histomorphology

The 10% NBF fixed ileal tissues were shipped to the Histology core facility, John A. Burns School of Medicine, UH Manoa, where samples were dehydrated with the series of ethanol solutions (70%, 80%, 95%, and 100%) and embedded in paraffin blocks. The embedded ileum tissue was sectioned at 6 μm thickness and stained with hematoxylin and eosin staining. A

total of 18 well oriented villus-crypts unit per sample was selected and observed under an 8× objective lens by using an Olympus microscope (U-TV0.63XC, Tokyo, Japan). Different intestinal morphological parameters such as villus height (VH)- distance from the tip of villus to the crypt, crypt depth (CD)- distance from villus base to the muscularis mucosa, and villus height to crypt depth ratio (VH/CD) [14] were measured along with the apparent villus surface area by using Infinite Analyze software (Lumenera Corporation, Ottawa, ON, Canada).

## Volatile fatty acid

The VFA analysis was carried out as previously described [15]. The VFA profile was analyzed by using gas chromatography (Trace 1300, Thermo Scientific, Waltham, MA) equipped with a flame ionization detector (FID), AS 1310 series automatic liquid sampler, and a 30 m × 0.53 mm internal diameter column (Stabilwax-DA, Restek, Restek Corporation, Bellefonte, PA). Helium was supplied as a carrier gas at a flow rate of 14.5 mL/min, and run time was set for 15 min. The temperature of the injector port and detector port was set at 200˚C and 240˚C, respectively. Trimethyl acetate (TMA) was used as an internal standard. Data handling and processing were performed on ChromeleonTM 7.2 software (Thermo Scientific, Waltham, MA).

## DNA extraction and 16S rRNA gene sequencing

Total genomic DNA was extracted from the cecal content using the QIAamp® DNA Stool Mini Kit (Qiagen, Hilden, Germany) following the manufacturer's instruction. The extracted bacterial DNA concentration was determined using NanoDrop one (Thermo Fisher Scientific, Madison, WI) and stored at -20˚C until further analysis. The V3 and V4 hypervariable regions of the 16S rRNA gene were amplified as outlined in Illumina 16S Metagenomic Sequencing Library guideline (Illumina) with the following modifications: Platinum Taq DNA Polymerase High Fidelity (Invitrogen, Life Technologies Corporation, Grand Island, NY) was used to set up the PCR reaction, and Mag-Bind Total Pure NGS beads (Omega Bio-Tek) were used for PCR Clean-Ups, and 35 cycles were used in the PCR. Briefly, 16S rRNA sequencing involved the following steps: 1) 1st stage PCR: Amplicon PCR, 2) PCR Clean-Up, 3) 2nd Stage PCR: Index PCR, 4) 2nd PCR Clean-Up 5) Library quantification, normalization, and pooling, and 6) Library denaturing and MiSeq sample loading. The specific sequence of 16S rRNA gene used for amplified were forward primer (5′–CCTACGGGNGGCWGCAG–3′) and reverse primer (5′GACTACHVGGGTATCTAATCC– 3′) [16]. Finally, amplicons were normalized and pooled, and subsequently sequenced on the Illumina MiSeq sequencer at the University of Hawaii at Manoa in Genomics, Proteomics, and Bioinformatics core facility.

## Sequencing processing

The Qiagen CLC Genomics Workbench 12.0.1 and the CLC Microbial Genomics module was used for microbial bioinformatics. The sequencing analysis procedures were followed as described in the OTU clustering step-by-step tutorial (Qiagen, Hilden, Germany). The sequencing files (ending with 'fastq') were imported in the CLC workbench; files were then paired, setting the minimum distance to 200 and maximum distance to 500. The reads were then trimmed, and the read with the low coverage was removed from the analysis. Thus, obtained reads were clustered as operational taxonomical units (OTUs), based on 97% sequence similarity against the Greengenes v13_8 97% database using the CLC Microbial Genomics module. Chimeras and OTUs with lower abundance (less than 10 reads) were removed. For alpha and beta diversity analysis, the phylogenetic tree was constructed using a maximum likelihood approach based on multiple sequence alignment (MSA) of the OTU

sequences generated by MUSCLE in the workbench. The Simpson's index and Shanon entropy were calculated to determine alpha diversity, while the unweighted UniFrac and weighted UniFrac distances were calculated to determine beta diversity. Permutational multivariate analysis of variance (PERMANOVA) analysis was carried out to measure the significance of beta diversity. Differences in microbiome among treatment groups were analyzed by one-way ANOVA to determine significant difference at different taxa level.

## Statistical analysis

Growth performance, gene expression, ileum histomorphology, and significantly abundant microbiome were analyzed using the R-studio and presented as mean ± SEM. These data were analyzed by using one-way analysis of variance (ANOVA), and mean separation between treatment groups was done by the Tukey-Kramer test. Kruskal-Wallis pairwise test and pairwise PERMANOVA test were carried out for alpha and beta diversity to determine the potential difference between treatment groups using the CLC Microbial Genomics module. The spearman correlation analysis was conducted to analyze associations between differential abundance taxonomic groups and measured change parameters using statistical software JMP v14 (SAS Institute Inc., Cary, NC). Values with P<0.05 were regarded as statistically significant.

## Results

### Effects of alpha-lipoic acid in growth performance

The growth performance of birds in different treatment groups is shown in Fig 1. Supplementation of the ALA significantly improved (P<0.05) the final bodyweight in the heat-stressed

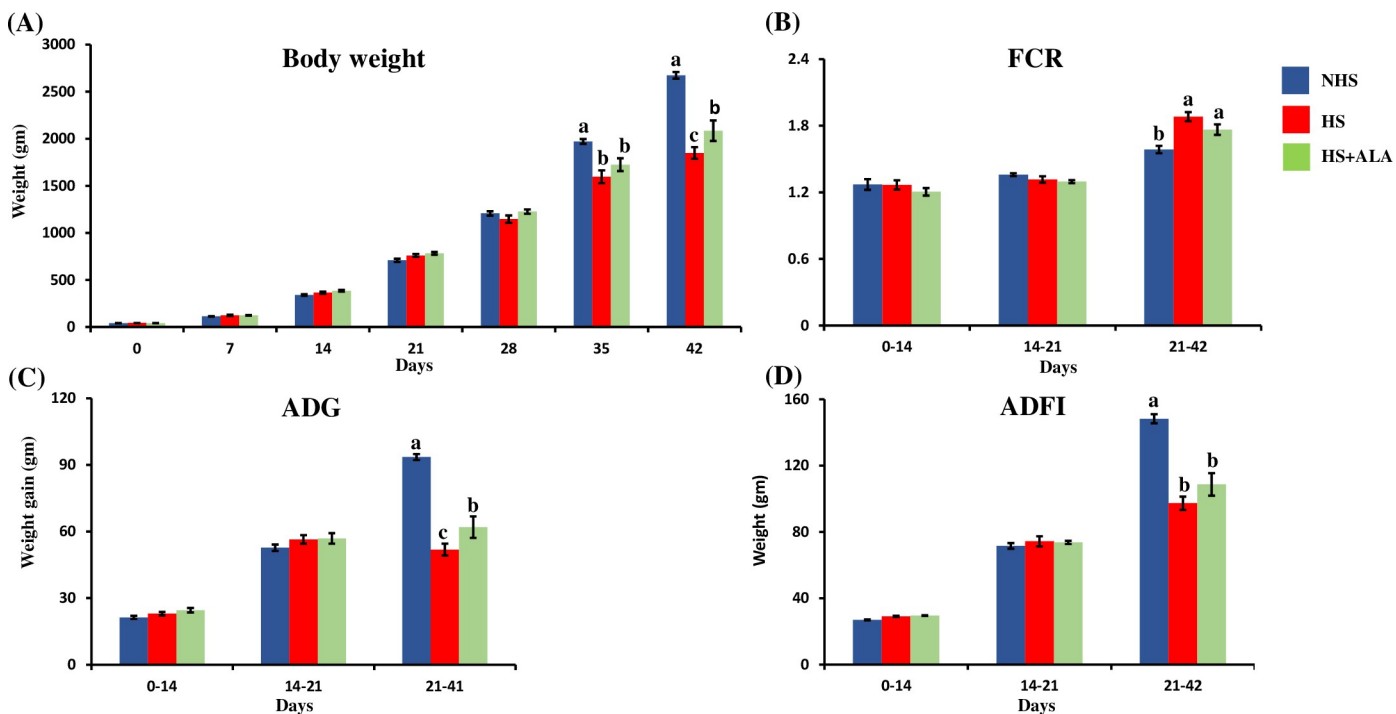

**Fig 1. Effects of ALA on the growth performance of heat-stressed broilers.** (A) Bodyweight, (B) FCR, (C) ADG, and (D) ADFI. Data presented as the mean± SEM (n = 24/treatment). The effect of treatment was statistically different at * P<0.05 for body weight, ADG, ADFI, and FCR. Different letters indicate the significant difference among the treatments.

broiler birds as compared to the HS group. There was no significant difference (P>0.05) in body weight between the treatment groups until 28 days. However, on 35 d, body weight was significantly decreased (P<0.05) in the HS group compared to the NHS group; dietary ALA supplementation in the heat-stressed broiler birds could not significantly improve the body weight. Nevertheless, on d 42, dietary ALA supplementation significantly improved (P<0.05) the bodyweight in the heat-stressed broiler birds. During the heat stress period (21–42 d), the ADFI was significantly lowered (P<0.05) in both HS and HS+ALA groups as compared to the NHS group. The ADG, on the other hand, was significantly lowered (P<0.05) in the HS group as compared to the NHS group, and dietary supplementation of ALA significantly improved (P<0.05) ADG in heat-stressed broiler birds during the heat stress period (21–42 d). FCR was significantly high (P<0.05) in the HS group and the ALA group compared to the NHS group. Dietary ALA supplementation was not able to significantly improve (P>0.05) the FCR in the heat-stressed broiler birds. There was no mortality in the NHS group, while one bird died in the HS group and the ALA group.

### Effects of ALA on the intestinal gene expression

**Expression of heat shock protein-related genes.** The mRNA expression of heat shock protein-related genes (*HSF1*, *HSF3*, *HSP70*, and *HSP90*) in the ileum of birds in different treatment groups is shown in Fig 2(A). The mRNA expression of *HSF1*, *HSF3*, and *HSP70* remains unchanged across the treatment groups. The mRNA expression of *HSP90* was significantly decreased (P<0.05) in the HS group as compared to the NHS group, while dietary ALA supplementation significantly increased (P<0.05) the mRNA expression of *HSP90* in heat-stressed broiler birds.

**Expression of tight-junction related genes.** The mRNA expression of the tight-junction related genes (*OCLN* and *CLDN1*) in the ileum of birds in different treatment groups is shown

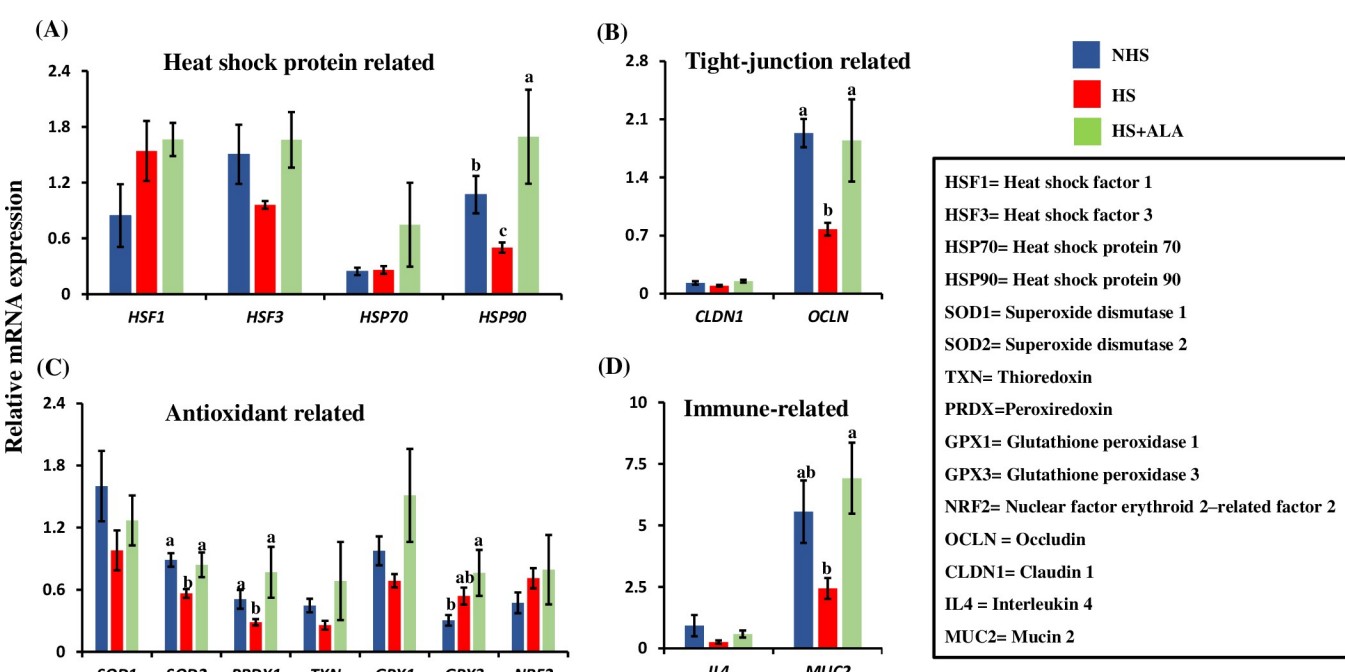

**Fig 2. Effects of ALA supplementation on the ileal gene expression of heat-stressed broilers.** (A) heat sock protein, (B) tight-junction related, (C) antioxidants related, and (D) immune-related genes. Data presented as the mean ±SEM (n = 6/treatment). Different letters indicate the significant difference among the treatments (* P<0.05).

in Fig 2(B). The heat stress significantly decreased (P<0.05) the mRNA expression of *OCLN* in the HS group as compared to the NHS group, while dietary ALA supplementation significantly increased (P<0.05) the expression of *OCLN* in heat-stressed broiler birds. The mRNA expression of *CLDN1* remains unchanged between treatment groups.

**Expression of antioxidant related genes.** The mRNA expression of the antioxidant-related genes (*SOD1*, *SOD2*, *PRDX1*, *TXN*, *GPX1*, *GPX3*, and *NRF2*) in the ileum of birds in different treatment groups are shown in Fig 2(C). The mRNA expression of the *SOD2* and *PRDX1* was significantly decreased (P<0.05) in the HS group as compared to the NHS group, while ALA supplementation significantly increased (P<0.05) their expressions in heat-stressed birds. Dietary ALA supplementation in the heat-stressed birds significantly increased (P<0.05) the mRNA expression of *GPX3* as compared to the NHS group. The mRNA expression of the *SOD1*, *TXN*, *GPX1*, and *NRF2* remain unchanged between the treatment groups.

**Expression of immune-related genes.** The mRNA expression of the immune-related genes (*IL4* and *MUC2*) between different treatment groups is shown in Fig 2(D). Dietary ALA supplementation in the heat-stressed broiler birds significantly increased (P<0.05) the expression of *MUC2* in heat-stressed broiler birds, as compared to the heat-stressed birds provided with just basal diet (HS group). The mRNA expression of the *IL4* remained unchanged between the treatment groups.

## Ileum histomorphology

The ileum histomorphology of birds in different treatment groups is shown in Fig 3. Villus height and Villus height to crypt depth ratio were significantly lowered (P<0.05) in the HS group as compared to the NHS group, while ALA supplementation significantly improved (P>0.05) these parameters in heat-stressed broiler birds. The villus surface area was significantly decreased (P<0.05) in the heat-stressed birds compared to the NHS group. Dietary ALA supplementation did not improve (P>0.05) the surface area in heat-stressed broiler birds. The crypt depth remained unchanged across the treatment groups.

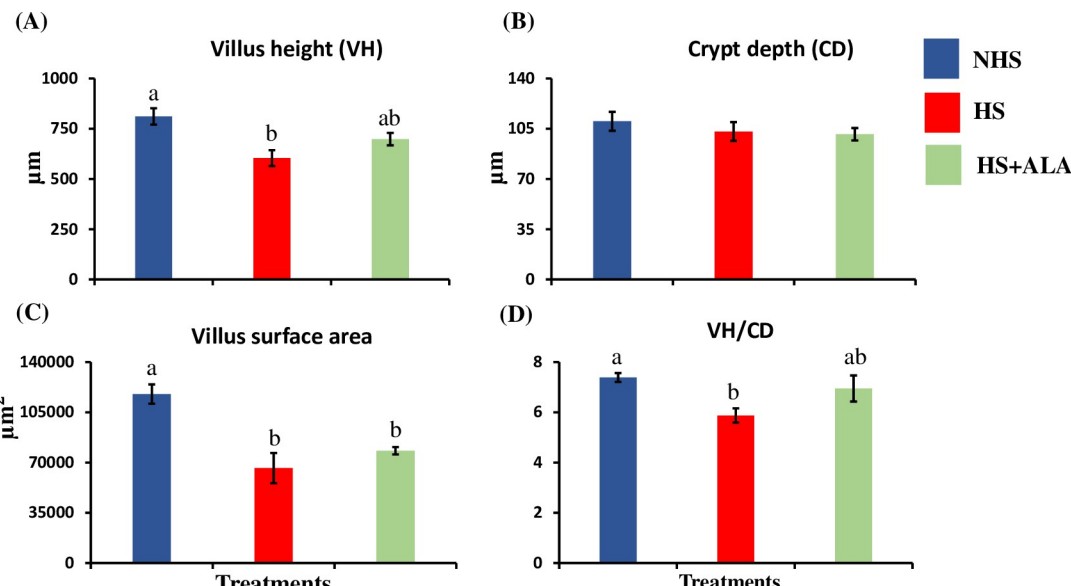

**Fig 3. Effects of ALA supplementation on the ileum histomorphology of the heat-stressed broilers.** (A) Villus height (VH), (B) Crypt depth (CD), (C) Villus surface area, (D) Villus height (VH): Crypt depth (CD). Data presented as the mean ±SEM (n = 4/treatment). The effect of treatment was statistically different at * P<0.05. Different letters indicate the significant difference among the treatments.

## Volatile fatty acids

The major VFA in the cecal digesta of birds in different treatment groups is shown in Fig 4. The amount of propionate was significantly decreased (P<0.05) in the HS groups as compared to the NHS groups. Simultaneously, dietary ALA supplementation significantly increased (P<0.05) propionate concentration in the cecal digesta compared to the HS group. The amount of acetate was significantly decreased (P<0.05) in the HS group as compared to the NHS group, while dietary ALA supplementation in heat-stressed broiler birds significantly increased its amount as compared to the HS group. Dietary ALA supplementation in the heat-stressed broiler birds also significantly increased (P<0.05) the concentration of butyrate as compared to the other two groups. Overall, total VFA was significantly lower (P<0.05) in the HS group as compared to the NHS group, and dietary ALA supplementation significantly increased (P<0.05) the concentration of the total VFA.

## Alpha and beta diversity of cecal microbiota

Shannon entropy was significantly higher (P<0.05) in the HS+ALA group as compared to the NHS group, while no differences (P>0.05) were observed in the Simpson index between these two groups (Fig 5). The unweighted UniFrac based PCoA revealed that bacterial composition in cecal content was significantly different (PERMANOVA analysis, P value = 0.01687) between the treatments, while no difference was observed in weighted UniFrac (Fig 6).

## Cecal microbial composition

At the phylum level, HS and HS+ALA groups were dominated by *Firmicutes* (63%, 63%), followed by *Bacteroidetes* (36%, 35%), while the NHS group has an almost equal abundance of *Firmicutes* (49%) and *Bacteroidetes* (50%) (Fig 7(A)). At the class level, the taxon-based analysis revealed that Clostridia and Bacteroidia were dominant classes across different treatment groups. However, these dominant taxa were not significantly different (P>0.05) across

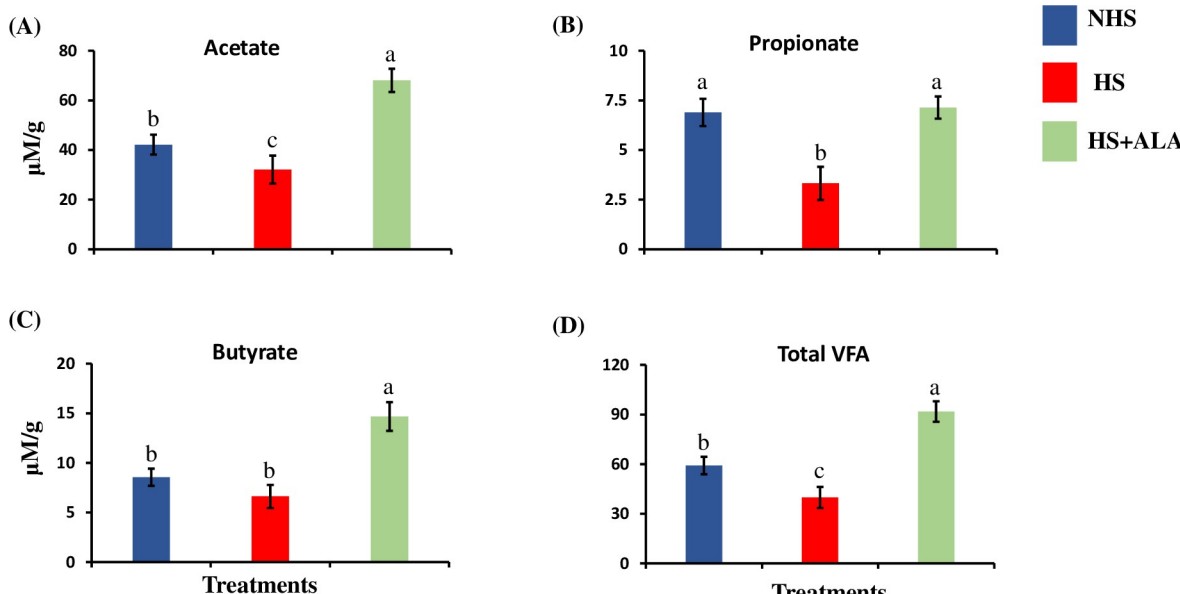

**Fig 4. Effects of ALA supplementation on the major volatile fatty acids in the cecal digesta of the heat-stressed broilers.** (A) Acetate, (B) Propionate, (C) Butyrate, and (D) Total VFA. Data presented as the mean ±SEM (n = 12/treatment). The effect of treatment was statistically different at * P<0.05. Different letters indicate the significant difference among the treatments.

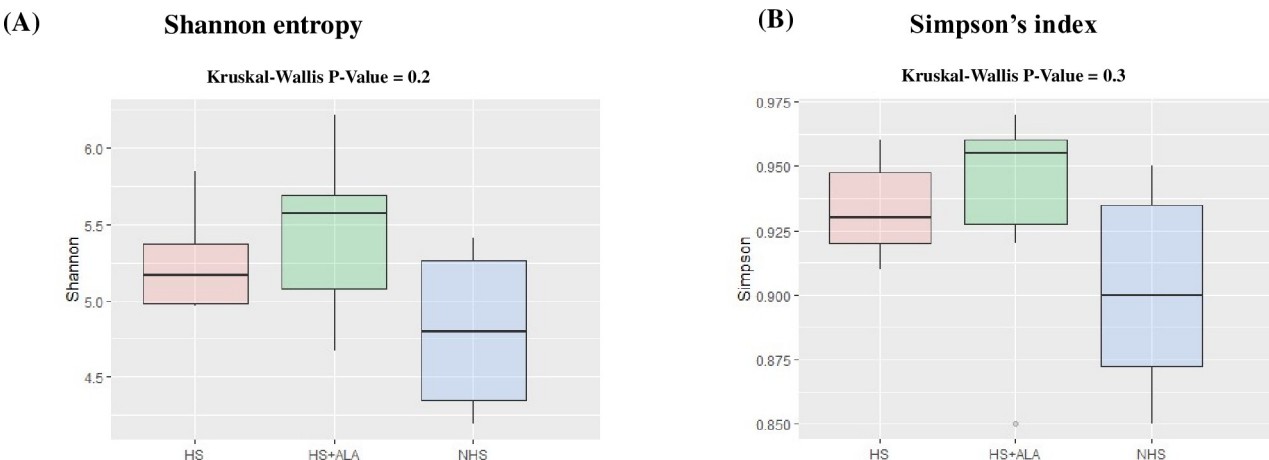

**Fig 5. Box plot showing the effects of ALA supplementation on microbial alpha diversity in heat-stressed broilers.** (A) Shannon entropy, and (B) Simpson's index.

treatment groups (Fig 7(B)). At the order level, *Clostridales* and *Bacteroidales* were dominant in different treatment groups (Fig 7(C)). These dominant orders were not statistically different between treatments. Dietary ALA supplementation, however, significantly increased (P<0.05) *Lactobacillales* in the heat-stressed birds (Fig 7(D)) as compared to the HS group.

The *Porphyromonadaceae*, *Ruminococcaceae*, *Lachnospiraceae*, and *unknown Family of Clostridiales* were the dominant family found across the treatment groups. The families *Lactobacillaceae* and *Peptostreptococcaceae* were significantly enriched (P<0.05) in the HS+ALA group compared to other groups (Fig 8).

At the genus level, *Parabacteroides* was the most dominant genus across all groups- HS (29%), HS+LA (24%), and NHS (44%), followed by *unknown family_Clostridales*. The relative abundance of these dominant genera was not significantly different among the groups. Dietary ALA supplementation, however, significantly enriched (P<0.05) the *Lactobacillus* and unknown genus of *Peptostreptococcaceae* in the heat-stressed birds (Fig 9).

## Correlation between the differential microbial species and measured parameters

The significant correlations observed after performing Spearman correlation analysis are shown in Table 3. The results showed that the expression of *SOD2* and *OCLN* were positively

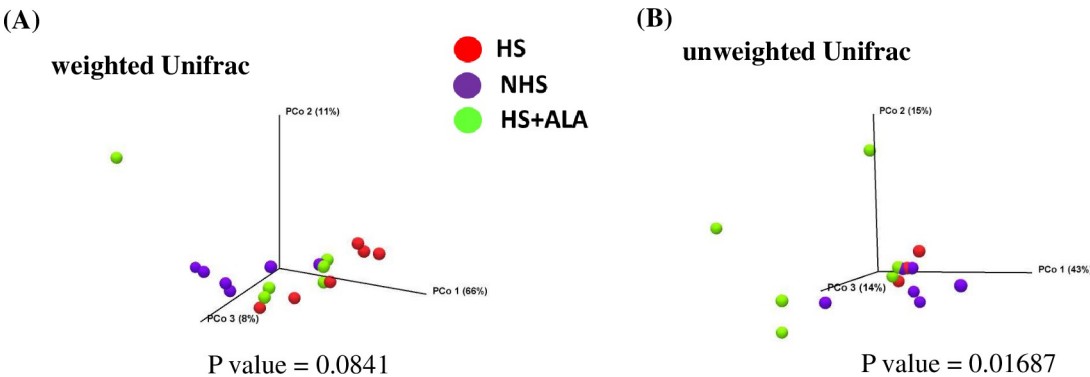

**Fig 6. Effects of ALA supplementation on microbial beta diversity in heat-stressed broilers.** (A) weighted UniFrac, and (B) unweighted UniFrac.

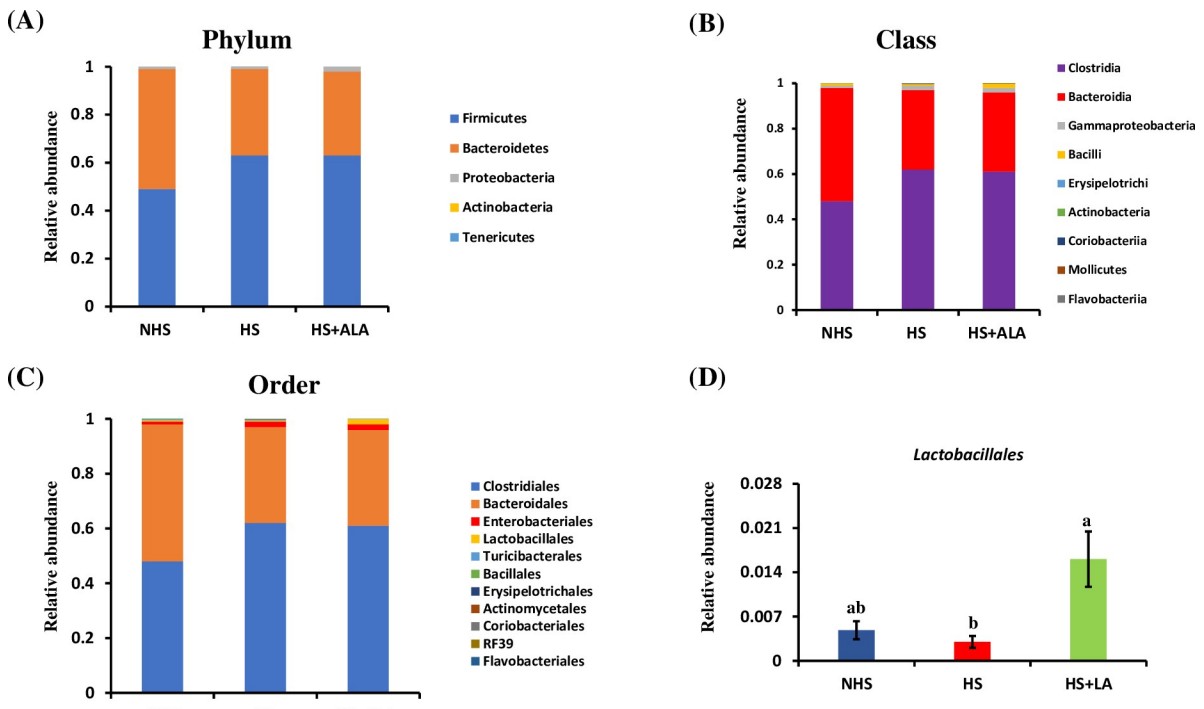

**Fig 7. Cecal microbial compositions.** (A) Relative abundance at Phylum level, (B) Relative abundance at Class level, (C) Relative abundance at the Order level, and (D) Significantly abundance of *Lactobacillales*.

(P<0.05) associated with order *Lactobacillales*. Acetate, butyrate, and total VFAs were positively correlated (P<0.05) with order *Lactobacillales*, family *Lactobacillaceae*, family *Peptostreptococcaceae*, genus *Lactobacillus*, and *unknown_genus_Peptostreptococcaceae*; While propionate was positively correlated (P<0.05) only with order *Lactobacillales*.

## Discussion

This study was carried out to evaluate the effectiveness of ALA (500 mg/kg of feed) to mitigate the negative effects of heat stress in poultry by observing growth performance along with different GI physiological parameters. Dietary supplementation of the ALA (500 mg/kg) significantly improved the final body weight, ADG, expressions of antioxidant, heat-shock, tight-junction, and immune-related genes, increased concentration of VFAs, and enriched beneficial gut microbiota in the heat-stressed broiler birds.

Heat stress negatively affects the production performance of broilers by decreased feed intake and growth [2, 17]. We also observed a decreased growth performance of broilers under heat stress. Several nutritional strategies are employed to mitigate the negative effects of heat stress in poultry. Among them, dietary supplementation of ALA as potential feed additives during heat stress has gained popularity in recent years and is found effective in mitigating the negative effects of heat stress in poultry [9]. In accord with those, we also observed significant improvement in the final body weight and ADG in heat-stressed broiler birds supplemented with ALA.

Further, we were interested in delineating the underlying GI physiological changes, leading to improved growth performance in heat-stressed birds supplemented with ALA. The intestine is an important organ that helps in nutrient digestion and absorption. High environmental temperature is found to have detrimental effects on the intestine [18]. Amongst different

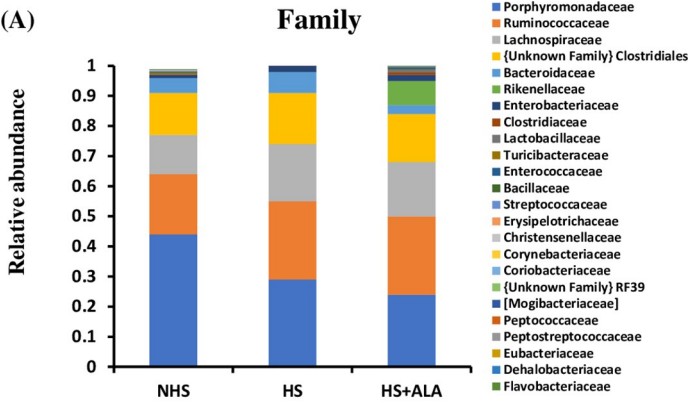

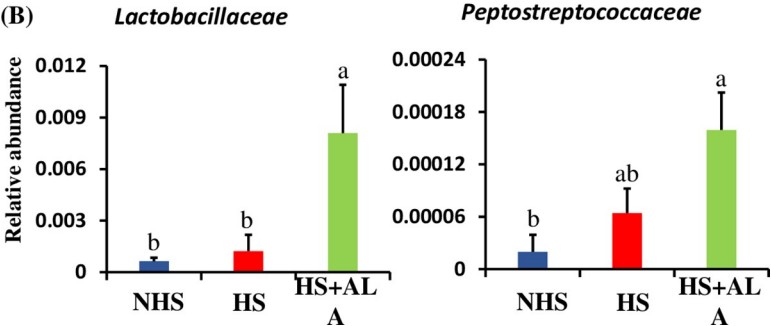

**Fig 8. Cecal microbial composition at the family level. (A) Relative abundance of microbiota, and (B) Significantly abundance microbiota within the family level.**

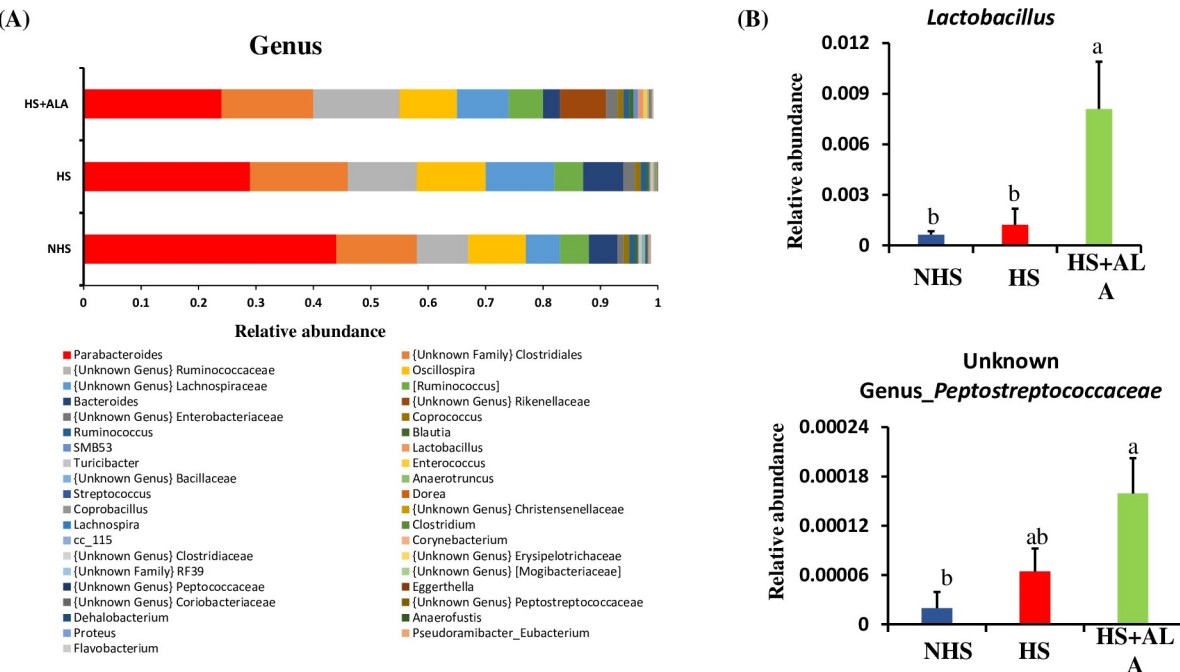

**Fig 9. Cecal microbial composition at the genus level. (A) Relative average abundance of microbiota, and (B) significantly abundance microbiota at the genus level.**

**Table 3. Spearman correlation between the differential microbial species and changed measure parameters.**

| Variables | Differential enriched microbes | Spearman ρ | Prob>\|ρ |
|---|---|---|---|
| SOD2 | O_Lactobacillales | 0.5119 | 0.0299* |
| OCLN | O_Lactobacillales | 0.4891 | 0.0394* |
| Acetate | O_Lactobacillales | 0.5595 | 0.0158* |
| Acetate | F_Lactobacillaceae | 0.7004 | 0.0012* |
| Acetate | F_Peptostreptococcaceae | 0.5199 | 0.0270* |
| Acetate | G_Lactobacillus | 0.7004 | 0.0012* |
| Acetate | G_Unknown_Genus_Peptostreptococcaceae | 0.5199 | 0.0270* |
| Propionate | O_Lactobacillales | 0.4871 | 0.0404* |
| Butyrate | O_Lactobacillales | 0.5429 | 0.0199* |
| Butyrate | F_Lactobacillaceae | 0.6715 | 0.0023* |
| Butyrate | F_Peptostreptococcaceae | 0.4795 | 0.0441* |
| Butyrate | G_Lactobacillus | 0.6715 | 0.0023* |
| Butyrate | G_Unknown_Genus_Peptostreptococcaceae | 0.4795 | 0.0441* |
| Total VFA | O_Lactobacillales | 0.5905 | 0.0099* |
| Total VFA | F_Lactobacillaceae | 0.7128 | 0.0009* |
| Total VFA | F_Peptostreptococcaceae | 0.471 | 0.0485* |
| Total VFA | G_Lactobacillus | 0.7128 | 0.0009* |
| Total VFA | G_Unknown_Genus_Peptostreptococcaceae | 0.471 | 0.0485* |

intestine parts, the ileum, a terminal part of the small intestine, is mainly susceptible to heat stress [19]. Different groups of genes (heat shock protein, antioxidant, tight junction, and immune-related genes) were then analyzed to detail the effects of ALA on heat-stressed broiler birds.

Heat shock proteins (HSPs)–chaperons–are the proteins that help properly fold the protein during the stress condition and possess cytoprotective action [20]. Different HSPs and their transcriptional factors were analyzed to determine the protective action of ALA during stress. Unlike previous studies [19, 21], in acute stress, where expression of HSP90 was higher in heat-stressed birds, expression HSP90 was lowered in this study. There have been varied expressions of HSFs and HSPs regarding tissue sample and the stress duration (i.e., acute vs. chronic stress) [22]. The spatiotemporal expression study of the HSPs and HSFs is therefore warranted. The lower expression of these genes in our study may be due to the exhaustion of the protective mechanism within the cells during chronic stress. Interestingly, dietary supplementation of ALA significantly increased the expression of HSP90.

Antioxidants are the molecules that scavenge the free radicals produced inside the cell. Heat stress elicits oxidative stress resulting in excess production of free radicals within the cell [23]. Therefore, different antioxidant-related genes were analyzed in this study to determine the effectiveness of ALA at the cellular level. The expressions of SOD2, PRDX1, and GPX3 were significantly improved in the heat-stressed broiler birds supplemented with ALA. SOD2 is mitochondrial manganese (Mn) containing enzymes that dismutase the superoxide radicals into hydrogen peroxide [24]. GPX3 reduces hydroperoxides and $H_2O_2$ by glutathione [25], while PRDX1 reduces $H_2O_2$ and hydroperoxides by using thioredoxin [26]. Like this study, previous studies have also demonstrated beneficial effects of ALA in improving total antioxidant capacity and antioxidant enzyme activity in oxidative stress conditions in broilers [27, 28].

The expression of tight junction protein and immune-related genes was then analyzed. The OCLN, a transmembrane protein, is one of the major tight junction protein that regulates

paracellular permeability and also plays a role in barrier functions [29]. Heat stress impairs the tight-junction protein in the intestine by reducing the blood flow and ultimately generating hypoxia in the intestine [30]. In light of that, the *OCLN* was lowered in the heat-stressed broiler birds in this study, while ALA supplementation improved the expression of *OCLN* in heat-stressed broiler birds. Besides, *MUC2* expression was also improved in ALA supplemented heat-stressed broiler birds. Mucin is essential in protecting the gut from pathogens, acidic environment, digestive enzymes, and nutrient digestion and absorption [31]. These results indicate that ALA supplementation improved the intestinal membrane integrity and protected the gut from pathogen and acidic environment, which might be one reason for improving the body weight and ADG in heat-stressed birds.

Heat stress impairs the intestinal morphology, decreases villus height, and villus height to crypt depth ratio [32–34]. In this study, heat stress altered the villus' morphology in the ileum, which may be due to intestinal ischemia, leading to epithelial shedding [35]. Although not statistically significant, the dietary ALA supplementation improved villus height and villus height to crypt depth ratio in heat-stressed birds. The improvement observed in the villus structure was possibly due to the VFAs produced in the gut as VFAs are exerting trophic effects on the intestinal morphology [36]. This observation drove us to analyze the major VFAs produced in the gut of heat-stressed birds. All major VFAs (acetate, propionate, and butyrate) concentrations were significantly increased in heat-stressed broiler birds supplemented with ALA. The improvement in the VFAs in heat-stressed birds was possibly due to the change in the microbiota as VFA is the microbial fermentation product [37]. Thus, we directed our study towards analyzing the microbiota changes in heat-stressed birds supplemented with ALA.

Dietary ALA supplementation significantly improved the relative abundance of *Lactobacillales* (order), *Lactobacillaceace* (family), *Peptostreptococcaceae* (family), *Lactobacillus*, and unknown genus of *Peptostreptococcaceae* in the heat-stressed broiler birds.

*Peptostreptococcaceae* belong to the phylum *Firmicutes* and play a role in gut homeostasis [38]. They are also found to produce VFAs in the intestine [39]. Spearman's correlation analysis in this study too revealed a positive association between *Peptostreptococcaceae* and acetate, butyrate, and total VFAs. *Lactobacillus* is the gram-positive bacteria that produce lactic acid, which reduces the pH and prevents the growth of pathogenic bacteria [40]. Thus, significant dominance of the *Lactobacillus* probably reduced the risk of pathogen amplification and invasion in heat-stressed birds supplemented with ALA. Besides, *lactobacillus* is found to have antioxidant capacity and remove the ROS to mitigate the damage induced by oxidative stress [41]. Moreover, *Lactobacillus* is also considered as a potent probiotic to improve gut health [42, 43]. Additionally, a positive association was observed between *Lactobacillus* and different gut health parameters such as expressions of *SOD2* and *OCLN*; and concentration of acetate, propionate, butyrate, and total VFAs. Considering these results, improved body weight and ADG in heat-stressed broilers supplemented with the ALA can be mainly attributed to improved gut microbiota and VFAs. More specifically, *Lactobacillus* and *Peptostreptococcaceae* might have played a vital role in improving growth performance in heat-stressed birds.

The alteration of the microbiota on supplementation of ALA can be attributed to the difference in acquisition and use of the lipoic acid among the different microbial species. There is diversity in the lipoate metabolism among different microbial species. One such example is a difference in lipoate metabolism between *Helicobacter pylori* and *Pseudomonas aeruginosa*. Protein encoding the lipoate metabolism is absent in the *H. pylori*, while *P. aeruginosa* contains the genome that encodes both the lipoate synthesis and lipoate scavenging enzymes as well as the components of the five lipoate complexes [44].

## Conclusion

The results of this study revealed that the dietary supplementation of ALA (500 mg/kg) significantly improved the final body weight and ADG; improved expressions of antioxidant, heat shock, immune-related, and tight-junction related genes in the ileum; increased concentration of VFAs and enriched the beneficial bacteria in the gut of heat-stressed broilers. Thus, ALA supplementation can be considered as one of the potential strategies to mitigate heat stress in broilers.

## Supporting information

**S1 Data.**
(XLSX)

## Acknowledgments

We sincerely thank Socorro Tauyan for helping in animal experimentation, Dr. Amit Singh in sampling, and Dr. Mohammad Arif for providing the facilities for Microbiome Analysis.

## Author Contributions

**Conceptualization:** Birendra Mishra.

**Data curation:** Sanjeev Wasti.

**Formal analysis:** Sanjeev Wasti, Nirvay Sah, Rajesh Jha, Birendra Mishra.

**Funding acquisition:** Birendra Mishra.

**Investigation:** Sanjeev Wasti, Birendra Mishra.

**Methodology:** Sanjeev Wasti, Nirvay Sah, Rajesh Jha, Birendra Mishra.

**Project administration:** Birendra Mishra.

**Supervision:** Birendra Mishra.

**Validation:** Chin N. Lee, Rajesh Jha.

**Writing – original draft:** Sanjeev Wasti, Nirvay Sah.

**Writing – review & editing:** Chin N. Lee, Rajesh Jha, Birendra Mishra.

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
