## [Decision Letter · Decision Letter 0]

9 Apr 2021

PONE-D-21-05713

Dietary Supplementation of Alpha-lipoic Acid Mitigates the Negative Effects of Heat Stress in Poultry

PLOS ONE

Dear Dr. Mishra,

Thank you for submitting your manuscript to PLOS ONE. After careful consideration, we feel that it has merit but does not fully meet PLOS ONE’s publication criteria as it currently stands. Therefore, we invite you to submit a revised version of the manuscript that addresses the points raised during the review process.

The manuscript should be revised deeply. The main problem found in the manuscript is related to the some aspects of methodology and redaction style. The manuscript should be presented according to guidelines for authors of Plos One. Please review the referee comments and make your peer revision. Thanks for your hard work.

We look forward to receiving your revised manuscript.

Kind regards,

Arda Yildirim, Ph.D.

Academic Editor

PLOS ONE

https://www.researchgate.net/profile/Arda_Yildirim2

Additional Editor Comments:

Firstly, thank you for your suggestion as an academic editor. This MS deals with an interesting and important topic in poultry production. Nevertheless there are still some points of concern from the reviewers, before the manuscript can be accepted for publication. Please make MS title specific because of focusing on animal material that is broiler. As there are some points unclear to me regarding the trial execution, sampling and statements in the MS, I recommend major revision.

Journal Requirements:

[This work was supported by a Start-up grant from CTAHR University of Hawaii at Manoa, and USDA Multistate (2052R) to B.M.Apart from providing funds, these organizations were not involved in any experimental procedure and manuscript preparation.]

 [N/A]

4. We noticed you have some minor occurrence of overlapping text with the following previous publication(s), which needs to be addressed:

- https://scholarspace.manoa.hawaii.edu/handle/10125/63507

Lines 184 - 194

In your revision ensure you cite all your sources (including your own works), and quote or rephrase any duplicated text outside the methods section. Further consideration is dependent on these concerns being addressed.

Reviewers' comments:

Reviewer's Responses to Questions

**Comments to the Author**

1. Is the manuscript technically sound, and do the data support the conclusions?

Reviewer #1: Yes

Reviewer #2: Yes

Reviewer #3: Partly

Reviewer #4: Yes

2. Has the statistical analysis been performed appropriately and rigorously? 

Reviewer #1: Yes

Reviewer #2: Yes

Reviewer #3: Yes

Reviewer #4: Yes

3. Have the authors made all data underlying the findings in their manuscript fully available?

Reviewer #1: Yes

Reviewer #2: Yes

Reviewer #3: Yes

Reviewer #4: Yes

4. Is the manuscript presented in an intelligible fashion and written in standard English?

Reviewer #1: Yes

Reviewer #2: Yes

Reviewer #3: Yes

Reviewer #4: Yes

5. Review Comments to the Author

Reviewer #1: The design and objectives of this study is interesting, and it is expected to contribute further to the knowledge base of improving broiler health by providing nutritional strategies to tackle heat-stress problem. The authors have done good work to make a comprehensive story of microbiota, immunity, heat stress parameters and performance. However, authors need to revise their work and correct some inherent concerns in the feed formulation. The other general comments and minor corrections are listed pointwise.

Major comments

1. What standard requirement was followed for this diet formulation? The calcium and phosphorus are extremely low.

The starter diet has high protein but comparatively low energy which is not as the right recommended ratio.

2. Were the birds weighed individually to show the standard deviation in each pen?

3. Confirm whether the gene expression calculation was based on relative standard curve or the comparative CT

method. Describe more specifically what represents the gene expression value.

4. Line 153. The citation for embedding is not properly attributed. The reviewer was not able to locate the information in

the cited article or in the reference attributed in the cited article! 

5. Since the authors have stated that ALA would have improved performance via microbiota-VFA modulation, it is

expected that they provide some insight into why ALA potentially modify microbiota.

Minor comments

Line 17: add ‘is’ between ‘which’ and ‘readily’

Line 20: State clearly the adaptation and allocation period along with the replication of the treatments.

Line 48: Better say heavier breast meat. Also, it should be improved breeding and not improved chicken.

Line 50: Insert ‘the’ between ‘and’ and ‘presence’.

Line 51: High temperature is the problem of tropical condition in the context of chicken.

Line 59: Check whether it will be is or are with ALA.

Line 82: Use n for experimental unit. Either state total number of birds per treatment or just mention birds per experimental unit.

Line 91: Remove ‘with’ after ‘fed’.

Line 110: Change ‘reducing’ to ‘subtracting’.

Line 158: Is it muscularis mucosae or submucosa?

Line 200-2001: Were the reads trimmed before pairing or after pairing?

Line 227: Add significance statistics.

Line 238: Remove numerical improvement. This means there was high variation among the replicates.

Line 244-245: Check grammar.

Line 262: Change ‘remains’ to ‘remain’.

Line 273: Use correct preposition after ‘compared’. Add significance statistics.

Line 288: Remove ‘total’ before ‘concentration’.

Line 298: Change ‘have’ to ‘had’.

Line 305: Add ‘the’ before ‘heat-stressed’.

Line 378: Is decreasing crypt depth an impairment?

Figure 5: Define the box plot. State whether the whiskers are error or data range. If the whiskers are range, then why are no data points on NHS box plot whisker?

Figure 7: What is the unit of relative abundance of Lactobacillales? Is it any fractions?

Reviewer #2: Dear Authors Regarding the manuscript title Dietary Supplementation of Alpha-lipoic Acid Mitigates the Negative Effects of Heat Stress in Poultry

The scientific background of the topic was well mentioned in the introduction part. The experiment design, as well as the replicates and methods used, were very good. The results obtained were presented in tables well discussed with other author’s results. However, some observation in the present paper should be corrected and add to improve the quality of the paper.

• The title (Dietary Supplementation of Alpha-lipoic A 1 cid Mitigates the Negative Effects of Heat Stress in Poultry) it will be better if you replacing Poultry to broilers

• Table 1 Ingredients and nutrient composition of the experimental diets, need to carful checking for the following :

1- SBM , add the crude protein level 44 or 46%?

2- MEn, kcal/kg, Ca, digP , not according to the Cobb-500 requirement.

• Introduction and Discussion

Need some other references about the impact of heat stress and Growth and Lipid Metabolism Marker Genes in Broiler Chickens in poultry I recommend you read the following references:

Saki Shimamoto 1,2, Kiriko Nakamura 1, Shozo Tomonaga 3, Satoru Furukawa 4, Akira Ohtsuka and Daichi Ijiri. Effects of Cyclic High Ambient Temperature and Dietary Supplementation of Orotic Acid, a Pyrimidine Precursor, on Plasma and Muscle Metabolites in Broiler Chickens. Metabolites 2020, 10, 189; doi:10.3390/metabo10050189

Saleh, Ahmed A.; Shukry, Mustafa; Farrag, Foad; Soliman, Mohamed M.; Abdel-Moneim, Abdel-Moneim E. (2021) "Effect of Feeding Wet Feed or Wet Feed Fermented by Bacillus licheniformis on Growth Performance, Histopathology and Growth and Lipid Metabolism Marker Genes in Broiler Chickens" Animals 11, no. 1: 83.

Ahmed A. Saleh , Mohammed S. Eltantawy , Esraa M. Gawish , Hassan H. Younis , Khairy A. Amber , Abd El-Moneim E. Abd El-Moneim & Tarek A. Ebeid (2020) Impact of Dietary Organic Mineral Supplementation on Reproductive Performance, Egg Quality Characteristics, Lipid Oxidation, Ovarian Follicular Development, and Immune Response in Laying Hens Under High Ambient Temperature. Biological Trace Element Research. 195:506–514.

Inoue H, Shimamoto S, Takahashi H, et al. Effects of astaxanthin-rich dried cell powder from Paracoccus carotinifaciens on carotenoid composition and lipid peroxidation in skeletal muscle of broiler chickens under thermo-neutral or realistic high temperature conditions. Anim Sci J. 2018;00:1–8.

Ahmed A. Saleh , Abeer A. Kirrella, Mahmoud A. O. Dawood, Tarek A. Ebeid (2019) Effect of dietary inclusion of cumin seed oil on the performance, egg quality, immune response and ovarian development in laying hens under high ambient temperature. Animal Physiology and Animal Nutrition. 103(6):1810-1817.

• Results: it will be better if you present the data for performance in table.

Reviewer #3: Heat stress is a practical problem to broiler chicks which results in over $128 million in losses for the poultry industry in the United States. The manuscript by Wasti et al examines whether using alpha lipoic acid (ALA) mitigated negative effects of heat stress in poultry chickens. The results document the beneficial effects of ALA in broiler chickens by improving gut health. The manuscript has been written in a well-orchestrated manner. However, some points have to be discussed and manuscript could be accepted only after major revisions.

General Comments:

Line 91: Remove the space error after “phases”, and also throughout the manuscript.

Line 97: Add space between “study[9]”

Line 111: No need to explain abbreviation ADG again once it already mentioned in the abstract section.

Line 238: Dietary ALA supplementation o was able to improve the FCR. What is supplementation o? Please remove it from text.

Specific Comments:

Comment 1: Figure 1,2,3,4 No word about sample size based on which the means and SEM were computed. Please add.

Comment 2: Most of the figures do not have the superscripts. Authors have to correct those figures.

Comment 3: The exact computed P-value at least for significant ones should be shown not only (P<0.05) which is a theoretical value.

Comment 4: Can authors please describe more for antioxidant defence system and its significance in poultry?

Comment 5: Please elaborate the ALA mechanism in improving growth performance of heat stress chickens in the discussion section.

Comment 6: Authors have not mentioned any dehydration-related parameters which is the most important factor to study the heat stress-related study as this causes dehydration in animals.

Comment 7: During dehydration, mainly epithelial cells hampered. Tight junction plays a major role during heat stress. The authors focused on the claudin and occludin molecules. Results showing that insignificant changes in claudin molecule but occludin is significantly higher in the treatment group. Tight junction maintains by both of these molecules. It is assuming from this result that due to heat stress, epithelial cells in the HS group individuals may experience dehydration condition which may lead to the loss of experimental animals. Is there so? If no, then how it will be justified? However, how it will be justified about the findings of villus height (higher in NHS), crypt depth (higher in NHS), and villus surface area (same in HS and ALA+HS) in relation to claudin and occludin level? Meantime, IL-4 is reduced which is an anti-inflammatory marker in the HS group.

Comment 8: Insert superscripts in Heat shock protein-related graph (Figure 2A). How it is possible the same level of HSP70 in the NHS and HS groups? It seems that the protein folding process will be the same in the control and HS group of animals. Then how HSP90 levels get reduced in the HS group?

Comment 9: Peroxiredoxin level showing different levels in the graph but it is showing the same superscripts? Why?

Comment 10: Most of the antioxidant-related parameter levels are higher in the treatment group. Meantime SOD1 and SOD2 are higher in the NHS group. How it will be justified?

Reviewer #4: It is a well-written manuscript. The study design and data collection are sufficient. The only concern is the number of animals used in the current study is low, and the growth performance data may not be representative. There are some minor comments:

Please indicate the sample number/statistical unit in each result. For example, if you collect two samples from each replicate. The data need to be averaged within each replicate upon analysis.

Is there any reason why ALA was not fed to the birds from the beginning of the trial?

It would be great if the author could present the morality from each treatment since it may impact the performance calculation due to low birds no.

Please also describe the raising environment, such as pen size and type (batter, cage or floor pen, litter, etc. )

6. PLOS authors have the option to publish the peer review history of their article (what does this mean?). If published, this will include your full peer review and any attached files.

Reviewer #1: No

Reviewer #2: No

Reviewer #3: **Yes: **Sahil Kalia

Reviewer #4: **Yes: **Chongxiao Chen

---

## [Author Response · Author response to Decision Letter 0]

19 May 2021

Reviewer #1: 

The design and objectives of this study is interesting, and it is expected to contribute further to the knowledge base of improving broiler health by providing nutritional strategies to tackle heat-stress problem. The authors have done good work to make a comprehensive story of microbiota, immunity, heat stress parameters and performance. However, authors need to revise their work and correct some inherent concerns in the feed formulation. The other general comments and minor corrections are listed pointwise.

Response: We highly appreciate the reviewer's thorough reading and insightful comments and suggestions on our manuscript. Based on the reviewer's suggestions, we have modified our manuscript, and highlighted it in “yellow” in the text of the manuscript.

Major comments

1. What standard requirement was followed for this diet formulation? The calcium and phosphorus are extremely low. The starter diet has high protein but comparatively low energy which is not as the right recommended ratio.

Response: Although official guidelines are still NRC 1994, we believe that is outdated. So, we formulate a ration based on the commercial birds' recommendation. Yet, mention as “meet or exceed the recommended requirements of NRC” (which is common among poultry researchers). In such a case, for starters, we target to have 21-22% CP and ~2900 Kcal/kg MEn diets. Also, there is debate on the Ca and P requirements with no any concrete recommendation. So, we are using a relatively lower dose of Ca and P in our formulation for broiler ration and go at the upper level in layer ration (to be safe). It seems that there was a little low in the finisher, though. However, we believe that it will not affect the result of this study, as it was similar in both the heat-stressed and control birds.

2. Were the birds weighed individually to show the standard deviation in each pen?

Response: Yes, birds were weighed individually in each pen.

3. Confirm whether the gene expression calculation was based on relative standard curve or the comparative CT method. Describe more specifically what represents the gene expression value.

Response: Gene expression calculation in our study was based on the comparative CT method. We confirmed the specificity of primers by running the melt curve, and gel electrophoresis for the specific product size.

4. Line 153. The citation for embedding is not properly attributed. The reviewer was not able to locate the information in the cited article or in the reference attributed in the cited article.

Response: Additional information has now been added, and the cited reference has been deleted from the text.

5. Since the authors have stated that ALA would have improved performance via microbiota-VFA modulation, it is expected that they provide some insight into why ALA potentially modify microbiota.

Response: Thank you for your suggestion. We have added explanation for the change in microbiota on supplementation of the VFA.

“The alteration of the microbiota on supplementation of ALA can be attributed to the difference in acquisition and use of the lipoic acid among the different microbial species. There is diversity in the lipoate metabolism among different microbial species. One such example is in between Helicobacter pylori and Pseudomonas aeruginosa. Protein encoding the lipoate metabolism is absent in the H. pylori, while P. aeruginosa contains the genome that encodes both the lipoate synthesis and lipoate scavenging enzymes as well as the components of the five lipoate complexes”.

Minor comments

Line 17: add ‘is’ between ‘which’ and ‘readily’

Response: Corrected

Line 20: State clearly the adaptation and allocation period along with the replication of the treatments.

Response: Additional information about the replication and the adaptation is now provided in the text (Line 21-22).

Line 48: Better say heavier breast meat. Also, it should be improved breeding and not improved chicken.

Response: We have made changes as per the reviewer’s suggestion.

Line 50: Insert ‘the’ between ‘and’ and ‘presence’.

Response: ‘the’ is now added in the text between ‘and’ and ‘presence’.

Line 51: High temperature is the problem of tropical condition in the context of chicken.

Response: We have now mentioned tropical regions in the text.

Line 59: Check whether it will be is or are with ALA.

Response: It is ‘is’.

Line 82: Use n for experimental unit. Either state total number of birds per treatment or just mention birds per experimental unit.

Response: It is now used in the text.

Line 91: Remove ‘with’ after ‘fed’.

Response: It is now removed from the text.

Line 110: Change ‘reducing’ to ‘subtracting’.

Response: It is now changed.

Line 158: Is it muscularis mucosae or submucosa?

Response: Crypt depth was measured from the villus base to the muscularis mucosa.

Line 200-2001: Were the reads trimmed before pairing or after pairing?

Response: It was paired and then trimmed.

Line 227: Add significance statistics.

Response: It is now added in the text.

Line 238: Remove numerical improvement. This means there was high variation among the replicates.

Response: The numerical improvement is now removed from the text.

Line 244-245: Check grammar.

Response: We have corrected the sentence.

Line 262: Change ‘remains’ to ‘remain’.

Response: The change is made in the text.

Line 273: Use correct preposition after ‘compared’. Add significance statistics.

Response: Correct preposition and statistical statistics is now added in the text.

Line 288: Remove ‘total’ before ‘concentration’.

Response: It is now removed from the text.

Line 298: Change ‘have’ to ‘had’.

Response: It is now changed in the text.

Line 305: Add ‘the’ before ‘heat-stressed’.

Response: It is now added in the text.

Line 378: Is decreasing crypt depth an impairment?

Response: It depends on the context. We have removed the word from the text.

Figure 5: Define the box plot. State whether the whiskers are error or data range. If the whiskers are range, then why are no data points on NHS box plot whisker?

Response: Thank you for the feedback. We have generated the new boxplot images using the R. In this boxplot; the whiskers represent the range of the data.

Figure 7: What is the unit of relative abundance of Lactobacillales? Is it any fractions?

Response: Yes, it is a fraction of Lactobacillales present in the particular treatment group.

Reviewer #2: 

Dear Authors Regarding the manuscript title Dietary Supplementation of Alpha-lipoic Acid Mitigates the Negative Effects of Heat Stress in Poultry. The scientific background of the topic was well mentioned in the introduction part. The experiment design, as well as the replicates and methods used, were very good. The results obtained were presented in tables well discussed with other author’s results. However, some observation in the present paper should be corrected and add to improve the quality of the paper.

Response: We highly appreciate the reviewer's thorough reading and insightful comments and suggestions on our manuscript. Based on the reviewer's suggestions, we have modified our manuscript, and highlighted it in “yellow” in the text of the manuscript.

• The title (Dietary Supplementation of Alpha-lipoic A 1 cid Mitigates the Negative Effects of Heat Stress in Poultry) it will be better if you replacing Poultry to broilers

Response: We have made changes in our title according to reviewer’s suggestion.

• Table 1 Ingredients and nutrient composition of the experimental diets, need to carful checking for the following :

1- SBM , add the crude protein level 44 or 46%?

2- MEn, kcal/kg, Ca, digP , not according to the Cobb-500 requirement.

Response: 1. We agreed with the reviewer that SBM is typically labeled as 44 to 48, depending on supplies. However, we used the analyzed value, which was 45% (although claimed 46%). So, we suggest not adding the protein value in the ingredient, like others, rather provide the nutrient content in the formulated diets. 

2. “Although official guidelines is still NRC 1994, we believe that is outdated. So, we formulate ration based on the commercial birds' recommendation. Yet, mention as “meet or exceed the recommended requirements of NRC” (which is common among poultry researchers). For starters, we target to have 21-22% CP and ~2900 Kcal/kg MEn diets. Also, there is debate on the Ca and P requirements with no any concrete recommendation. So, we are using a relatively lower dose of Ca and P in our formulation for broiler ration and go at the upper level in layer ration (to be safe). It seems that there was a little low in the finisher, though. However, we believe that it will not affect the result of this study, as it was similar in both heat-stressed and normal burds group.”

• Introduction and Discussion

Need some other references about the impact of heat stress and Growth and Lipid Metabolism Marker Genes in Broiler Chickens in poultry I recommend you read the following references:

Response: We have now included the following references in our manuscript.

• Saki Shimamoto 1,2, Kiriko Nakamura 1, Shozo Tomonaga 3, Satoru Furukawa 4, Akira Ohtsuka and Daichi Ijiri. Effects of Cyclic High Ambient Temperature and Dietary Supplementation of Orotic Acid, a Pyrimidine Precursor, on Plasma and Muscle Metabolites in Broiler Chickens. Metabolites 2020, 10, 189; doi:10.3390/metabo10050189

• Saleh, Ahmed A.; Shukry, Mustafa; Farrag, Foad; Soliman, Mohamed M.; Abdel-Moneim, Abdel-Moneim E. (2021) "Effect of Feeding Wet Feed or Wet Feed Fermented by Bacillus licheniformis on Growth Performance, Histopathology and Growth and Lipid Metabolism Marker Genes in Broiler Chickens" Animals 11, no. 1: 83.

• Ahmed A. Saleh , Mohammed S. Eltantawy , Esraa M. Gawish , Hassan H. Younis , Khairy A. Amber , Abd El-Moneim E. Abd El-Moneim & Tarek A. Ebeid (2020) Impact of Dietary Organic Mineral Supplementation on Reproductive Performance, Egg Quality Characteristics, Lipid Oxidation, Ovarian Follicular Development, and Immune Response in Laying Hens Under High Ambient Temperature. Biological Trace Element Research. 195:506–514.

• Inoue H, Shimamoto S, Takahashi H, et al. Effects of astaxanthin-rich dried cell powder from Paracoccus carotinifaciens on carotenoid composition and lipid peroxidation in skeletal muscle of broiler chickens under thermo-neutral or realistic high temperature conditions. Anim Sci J. 2018;00:1–8.

• Ahmed A. Saleh , Abeer A. Kirrella, Mahmoud A. O. Dawood, Tarek A. Ebeid (2019) Effect of dietary inclusion of cumin seed oil on the performance, egg quality, immune response and ovarian development in laying hens under high ambient temperature. Animal Physiology and Animal Nutrition. 103(6):1810-1817.

• Results: it will be better if you present the data for performance in table.

Response: For better visibility of growth performance data, we presented the findings in bar graphs. A supplemental file is now submitted along with the manuscript, which includes performance data in a tabular format.

 

Reviewer #3:

Heat stress is a practical problem to broiler chicks which results in over $128 million in losses for the poultry industry in the United States. The manuscript by Wasti et al examines whether using alpha lipoic acid (ALA) mitigated negative effects of heat stress in poultry chickens. The results document the beneficial effects of ALA in broiler chickens by improving gut health. The manuscript has been written in a well-orchestrated manner. However, some points have to be discussed and manuscript could be accepted only after major revisions.

Response: We highly appreciate the reviewer's thorough reading and insightful comments and suggestions on our manuscript. Based on the reviewer's suggestions, we have modified our manuscript, and highlighted it in “yellow” in the text of the manuscript.

General Comments:

Line 91: Remove the space error after “phases”, and also throughout the manuscript.

Response: We have now corrected it throughout the manuscript. 

Line 97: Add space between “study[9]”

Response: Added 

Line 111: No need to explain abbreviation ADG again once it already mentioned in the abstract section.

Response: Corrected

Line 238: Dietary ALA supplementation o was able to improve the FCR. What is supplementation o? Please remove it from text.

Response: Corrected

Specific Comments:

Comment 1: Figure 1,2,3,4 No word about sample size based on which the means and SEM were computed. Please add.

Response: We appreciate the reviewer for pointing out this issue. Information about the sample size is now provided in the figure.

Comment 2: Most of the figures do not have the superscripts. Authors have to correct those figures.

Response: Thank for your suggestion. However, we have used the different letters/ superscripts only in the figures where there is statistically significance. We believe that will make the reader easily visualize the main findings of the figure, and there is no point in using the superscript in the figures where there is not statistically significant.

Comment 3: The exact computed P-value at least for significant ones should be shown not only (P<0.05) which is a theoretical value.

Response: Thank you for your suggestion. We have now included that information in the supplemental data provide along with this manuscript.

Comment 4: Can authors please describe more for antioxidant defense system and its significance in poultry?

Response: Thanks for your suggestion. Information about the antioxidant defense system is provided in the discussion section.

Comment 5: Please elaborate the ALA mechanism in improving growth performance of heat stress chickens in the discussion section.

Response: Besides observing the growth performance, the objective of this paper was to delineate the mechanism the beneficial aspects of the ALA in heat-stressed birds. We have tried to incorporate the mechanism by measuring different parameters – VFAs, gene expression, histomorphology and microbiota. All these aspects have been discussed in the discussion section. Moreover, the mechanism by which ALA helped in the improvement of microbiota is also now included in the manuscript.

Comment 6: Authors have not mentioned any dehydration-related parameters which is the most important factor to study the heat stress-related study as this causes dehydration in animals.

Response: Yes, we agreed with the reviewer’s point that we should have included dehydration-related parameters such as – water intake etc. We were not able to precisely collect those data because of our animal experimentation facility. All the birds in HS group were provided with ad libitum of water. We are, however, planning to include those parameters in our future experiments.

Comment 7: During dehydration, mainly epithelial cells hampered. Tight junction plays a major role during heat stress. The authors focused on the claudin and occludin molecules. Results showing that insignificant changes in claudin molecule but occludin is significantly higher in the treatment group. Tight junction maintains by both of these molecules. It is assuming from this result that due to heat stress, epithelial cells in the HS group individuals may experience dehydration condition which may lead to the loss of experimental animals. Is there so? If no, then how it will be justified? However, how it will be justified about the findings of villus height (higher in NHS), crypt depth (higher in NHS), and villus surface area (same in HS and ALA+HS) in relation to claudin and occludin level? Meantime, IL-4 is reduced which is an anti-inflammatory marker in the HS group.

Response: Yes, there was mortality. One of the birds died in the HS and the ALA group. Besides Occludin and Claudin, there are a lot of other molecules that affect the morphology of the villi in the intestine. Occludin and claudin are only a few tight-junction proteins chosen to observe the beneficial effect of ALA in heat-stressed birds. A comprehensive study needs to be undertaken if we want to delineate the exact mechanism of how HS is affecting the villus height and crypt depth at the molecular level. We won’t be able to come up with a concrete interpretation just by observing these two molecules.

 IL-4 has many biological roles inside the cell. It is involved in the T-cell proliferation and differentiation of B cells into the plasma cells. Besides, IL-4 also induces B-cell class switching to IgE. Taken together, decreased IL-4 level in the HS group shows the lower immunity of the birds during the high temperature. 

Comment 8: Insert superscripts in Heat shock protein-related graph (Figure 2A). How it is possible the same level of HSP70 in the NHS and HS groups? It seems that the protein folding process will be the same in the control and HS group of animals. Then how HSP90 levels get reduced in the HS group?

Response: Studies have revealed that the expression of these molecules varies with the intensity of the stress, i.e., either acute or chronic. In our study, we did not find a significant difference in the expression of HSP70 between the treatment groups. In our opinion, to justify the same level of expression of the HSP70 gene in between NHS and HS, spatiotemporal expression of this gene needs to be carried out, which can come up with any conclusive statements about its expression pattern. Moreover, in our follow-up study, we plan to study the spatiotemporal expression of these genes in heat-stressed birds.

Comment 9: Peroxiredoxin level showing different levels in the graph but it is showing the same superscripts? Why?

Response: It is so because of the error bars (variation). If you look at the error bars of NHS and HS+ALA groups, they coincide, which indicates that they were not statistically different. That is the reason the same superscript was given in between two groups – NHS and HS+ALA.

Comment 10: Most of the antioxidant-related parameter levels are higher in the treatment group. Meantime SOD1 and SOD2 are higher in the NHS group. How it will be justified?

Response: The higher level of the SOD1 and SOD2 in the NHS group observed in this study was in agreement with our expectation. The higher of SOD1 and SOD2 indicates the high amount of this antioxidant inside the cell; when we exposed the animal to the heat stress condition, the relative expression of these genes went down, which we expect in the heat stress. Again, on supplementing ALA, the level of these genes went up. These findings reflect the perfect antioxidant mechanism inside the stressed cell.

Reviewer #4: 

It is a well-written manuscript. The study design and data collection are sufficient. The only concern is the number of animals used in the current study is low, and the growth performance data may not be representative. There are some minor comments:

Response: We highly appreciate the reviewer's thorough reading and insightful comments and suggestions on our manuscript. Based on the reviewer's suggestions, we have modified our manuscript and highlighted it in “yellow” in the text of the manuscript.

Please indicate the sample number/statistical unit in each result. For example, if you collect two samples from each replicate. The data need to be averaged within each replicate upon analysis.

Response: The sample number is now mentioned below the figure in the result section.

Is there any reason why ALA was not fed to the birds from the beginning of the trial?

Response: The objective of this study was to observe the mitigatory effects of ALA in heat-stressed broilers. We exposed the birds to the heat stress condition only after 21 days, so we choose to provide them only during the heat stress period. Also, in a real-life practical scenario, farmers will prefer to supplement the products only at the HS time.

It would be great if the author could present the morality from each treatment since it may impact the performance calculation due to low birds no.

Response: We really appreciate your comment regarding mortality. One bird died from the HS group and the HS+ALA group. This information is now provided in the result section.

Please also describe the raising environment, such as pen size and type (batter, cage or floor pen, litter, etc. )

Response: The birds were raised on the floor pen system, the size of the individual pen was 1 m x 0.71 m, and the stocking density was 1500 cm2/bird. These informations are now added to the text.

---

## [Decision Letter · Decision Letter 1]

25 Jun 2021

PONE-D-21-05713R1

Dietary Supplementation of Alpha-lipoic Acid Mitigates the Negative Effects of Heat Stress in Broilers

PLOS ONE

Dear Dr. Mishra,

Thank you for submitting your manuscript to PLOS ONE. After careful consideration, we feel that it has merit but does not fully meet PLOS ONE’s publication criteria as it currently stands. Therefore, we invite you to submit a revised version of the manuscript that addresses the points raised during the review process.

Please provide clear information on whether the non-standard requirement of a commercial hybrid broiler for calcium and phosphorus (available) values is met.

We look forward to receiving your revised manuscript.

Kind regards,

Arda Yildirim, Ph.D.

Academic Editor

PLOS ONE

Journal Requirements:

Additional Editor Comments (if provided):

For your guidance, you can check the reviewers' comments.

Reviewers' comments:

Reviewer's Responses to Questions

**Comments to the Author**

1. If the authors have adequately addressed your comments raised in a previous round of review and you feel that this manuscript is now acceptable for publication, you may indicate that here to bypass the “Comments to the Author” section, enter your conflict of interest statement in the “Confidential to Editor” section, and submit your "Accept" recommendation.

Reviewer #1: (No Response)

Reviewer #2: All comments have been addressed

Reviewer #3: All comments have been addressed

2. Is the manuscript technically sound, and do the data support the conclusions?

Reviewer #1: Yes

Reviewer #2: Yes

Reviewer #3: Yes

3. Has the statistical analysis been performed appropriately and rigorously? 

Reviewer #1: Yes

Reviewer #2: Yes

Reviewer #3: Yes

4. Have the authors made all data underlying the findings in their manuscript fully available?

Reviewer #1: Yes

Reviewer #2: Yes

Reviewer #3: Yes

5. Is the manuscript presented in an intelligible fashion and written in standard English?

Reviewer #1: Yes

Reviewer #2: Yes

Reviewer #3: Yes

6. Review Comments to the Author

Reviewer #1: 1. More specifically, the calcium and phosphorus values are not formulated according to the commercial birds recommendation (in this case Cobb 500). So, the authors answer is not correct. It is true that there is variation in the inclusion level of calcium and phosphorus. The authors must mention in the manuscript that they formulated a low calcium and phosphorus diet and if possible, should provide the reason for such formulation. Low Ca and P do not necessarily keep the birds safe. This is not going to affect the outcome of this study but would provide a wrong reference for other researchers that low Ca and P diet would be required to yield a standard growth performance. The authors should provide correct information. They cannot cite one reference for meeting requirement and provide reason that it is outdated, and they considered commercial recommendation, surprisingly which was even not followed. Cobb guidelines state Ca requirement to be 0.9% in the starter and 0.8% in the finisher, available phosphorus is around 0.45%. The authors have formulated the diet containing 0.75% Ca in the starter and 0.52% in the finisher, while the available phosphorus is 0.3% in the starter and 0.23% in the finisher. There is a typo: it should be available phosphorus and not dig. If the authors provide correct information in the draft and state the reason for this inclusion level then it would be ok otherwise it would be understood as a mistake in their feed formulation.

2. Also, authors should have removed 'standard curve method' from line 146 if they calculated gene expression by the comparative CT method. Figure 2 does not show any fold change (as stated in line 147) as none of the treatment has a normalized value of 1. So, it would be better to remove the fold change from the materials and methods. Besides, the authors should mention the treatment (may be the control group) that was used to calculate the ΔΔ CT.

Reviewer #2: Dear Authors Regarding the manuscript title Dietary Supplementation of Alpha-lipoic Acid Mitigates the Negative Effects of Heat Stress in Broilers

The scientific background of the topic was well mentioned in the introduction part. The experiment design, as well as the replicates and methods used, were very good. The results obtained were presented in tables well discussed with other author’s results. In addition the authors answered all inquiries and the manuscript may accept in this form.

Reviewer #3: (No Response)

7. PLOS authors have the option to publish the peer review history of their article (what does this mean?). If published, this will include your full peer review and any attached files.

Reviewer #1: No

Reviewer #2: No

Reviewer #3: **Yes: **Sahil Kalia

---

## [Author Response · Author response to Decision Letter 1]

2 Jul 2021

Reviewer #1

More specifically, the calcium and phosphorus values are not formulated according to the commercial birds’ recommendation (in this case Cobb 500). So, the authors answer is not correct. It is true that there is variation in the inclusion level of calcium and phosphorus. The authors must mention in the manuscript that they formulated a low calcium and phosphorus diet and if possible, should provide the reason for such formulation. Low Ca and P do not necessarily keep the birds safe. This is not going to affect the outcome of this study but would provide a wrong reference for other researchers that low Ca and P diet would be required to yield a standard growth performance. The authors should provide correct information. They cannot cite one reference for meeting requirement and provide reason that it is outdated, and they considered commercial recommendation, surprisingly which was even not followed. Cobb guidelines state Ca requirement to be 0.9% in the starter and 0.8% in the finisher, available phosphorus is around 0.45%. The authors have formulated the diet containing 0.75% Ca in the starter and 0.52% in the finisher, while the available phosphorus is 0.3% in the starter and 0.23% in the finisher. There is a typo: it should be available phosphorus and not dig. If the authors provide correct information in the draft and state the reason for this inclusion level then it would be ok otherwise it would be understood as a mistake in their feed formulation.

Response: We highly appreciate the reviewer’s meticulous review and pointing out about the feed formulation (following the recent commercial standard). Yes, we agree with the reviewer’s point that we have used lower amount of the calcium and phosphorus than that is recommended for the commercial requirements. We have added this information in our revised manuscript. (Line 94-95). The objective of this study is focused on mitigating heat stress using nutritional supplementation (Alpha lipoic acid). So, our focus was on meeting the energy and protein requirement following the NRC (1994) guidelines and examining at the variables related to heat stress. We agree with the reviewer that we should have formulate diets based on commercial requirement, which we believe is a valuable input for executing our future projects. In this study, however, we believe that lower Ca and P level will not affect the tested variable. So, the finding of the study still holds scientific merits as it provides valuable information on mitigation of heat stress. 

Corrections:

• Birds were fed the corn-soybean meal-based diets in two phase, starter (1-21d) and finisher (22-42d). The energy and protein requirements of the diet was met following the NRC (1994); however, Ca and P level was used at lower level than the commercial requirements (Line 94-95). 

• We have corrected “dig-P” to “available phosphorous” (in the Table-1, Line 102)

Again, thank you so much for your critical review for the refinement of this manuscript, and we will follow the recent NRC guidelines in our future study as per reviewer’s recommendation. 

2. Also, authors should have removed 'standard curve method' from line 146 if they calculated gene expression by the comparative CT method. Figure 2 does not show any fold change (as stated in line 147) as none of the treatment has a normalized value of 1. So, it would be better to remove the fold change from the materials and methods. Besides, the authors should mention the treatment (may be the control group) that was used to calculate the ΔΔ CT.

Response: Standard curve has been removed from the method section, and was modified

In line 147-150: The expression level of target genes was determined using the cycle threshold (Ct) values and changes in the gene expression were calculated by the 2-ΔΔCt method compared to the control group. The relative mRNA expression was normalized to the endogenous reference gene B-actin.

---

## [Decision Letter · Decision Letter 2]

7 Jul 2021

Dietary Supplementation of Alpha-lipoic Acid Mitigates the Negative Effects of Heat Stress in Broilers

PONE-D-21-05713R2

Dear Dr. Mishra,

We’re pleased to inform you that your manuscript has been judged scientifically suitable for publication and will be formally accepted for publication once it meets all outstanding technical requirements.

Kind regards,

Arda Yildirim, Ph.D.

Academic Editor

PLOS ONE

Additional Editor Comments (optional):

Thanks for reviewing and accepting all the comments and suggestions.

Reviewers' comments:

Reviewer's Responses to Questions

**Comments to the Author**

1. If the authors have adequately addressed your comments raised in a previous round of review and you feel that this manuscript is now acceptable for publication, you may indicate that here to bypass the “Comments to the Author” section, enter your conflict of interest statement in the “Confidential to Editor” section, and submit your "Accept" recommendation.

Reviewer #1: All comments have been addressed

2. Is the manuscript technically sound, and do the data support the conclusions?

Reviewer #1: Yes

3. Has the statistical analysis been performed appropriately and rigorously? 

Reviewer #1: Yes

4. Have the authors made all data underlying the findings in their manuscript fully available?

Reviewer #1: Yes

5. Is the manuscript presented in an intelligible fashion and written in standard English?

Reviewer #1: Yes

6. Review Comments to the Author

Reviewer #1: The comments have been addressed. Congratulations to the authors for their hard work and acceptable draft for publication.

7. PLOS authors have the option to publish the peer review history of their article (what does this mean?). If published, this will include your full peer review and any attached files.

Reviewer #1: No

---

## [Editor Report · Acceptance letter]

16 Jul 2021

PONE-D-21-05713R2 

Dietary Supplementation of Alpha-lipoic Acid Mitigates the Negative Effects of Heat Stress in Broilers 

Dear Dr. Mishra:

I'm pleased to inform you that your manuscript has been deemed suitable for publication in PLOS ONE. Congratulations! Your manuscript is now with our production department. 

Kind regards, 

on behalf of

Prof. Dr. Arda Yildirim 

Academic Editor

PLOS ONE